# Numerical Computation-Based Position Estimation for QR Code Object Marker: Mathematical Model and Simulation

**Mooi Khee Teoh** [1] **, Kenneth T. K. Teo** [2] **and Hou Pin Yoong** [1,*]

1   Biomechatronics Research Laboratory, Faculty of Engineering, Universiti Malaysia Sabah, Jalan UMS, Kota Kinabalu 88400, Malaysia
2   Modelling, Simulation and Computing Laboratory, Faculty of Engineering, Universiti Malaysia Sabah, Jalan UMS, Kota Kinabalu 88400, Malaysia
*   Correspondence: yoongpin@ums.edu.my

**Abstract:** Providing position and orientation estimations from a two-dimensional (2D) image is challenging, as such images lack depth information between the target and the automation system. This paper proposes a numerical-based monocular positioning method to determine the position and orientation of a single quick response (QR) code object marker. The three-dimensional (3D) positional information can be extracted from the underdetermined system using the QR code's four vertices as positioning points. This method uses the fundamental principles of the pinhole imaging theory and similar triangular rules to correspond the QR code's corner points in a 3D environment to the 2D image. The numerical-based model developed with suitable guessing parameters and correct updating rules successfully determines the QR code marker's position. At the same time, an inversed rotation matrix determines the QR code marker's orientation. Then, the MATLAB platform simulates the proposed positioning model to identify the maximum rotation angles detectable at various locations using a single QR code image with the known QR code's size and the camera's focal length. The simulation results show that the proposed numerical model can measure the position and orientation of the tilted QR code marker within 30 iterations with great accuracy. Additionally, it can achieve no more than a two-degree angle calculation error and less than a five millimeter distance difference. Overall, more than 77.28% of the coordinate plane simulated shows a converged result. The simulation results are verified using the input value, and the method is also capable of experimental verification using a monocular camera system and QR code as the landmark.

**Keywords:** numerical method; 3D positioning; pose estimation; pinhole imaging; machine vision; QR code marker

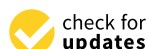



## 1. Introduction

The COVID-19 pandemic, with its associated transmission risk and infection concerns, as well as the resulting travel restrictions and movement controls, has led to a surge in online grocery ordering and delivery services. Consequently, pick-cart jobs have become overloaded, as online grocery purchasing necessitates the grocer to retrieve items from their inventory, pack them, and deliver them to the buyer. The implementation of an unmanned pick-cart robot would be extremely useful in such situations. One of the most vital aspects when designing an autonomous robot is identifying the target's position and orientation before calculating the gripper's path to reach the object [1–3]. The use of a monocular vision positioning system with artificial markers can assure the system's performance. Quick response (QR) codes are printed on the grocery packaging, and they can be used as artificial markers. Thus, this eliminates the need to design markers and place them manually on each item. This study is crucial, as the proposed iterative-based pose estimation model provides a new approach to determining the three-dimensional (3D) position of a single tilted QR code from an image captured at a single location using a monocular camera.

Generally, imaging sensing systems use multiple cameras for localization, as can be seen for the widely used positioning systems used in landmark-based localization and navigation (L&N) systems [4–7] and augmented reality technologies [8]. In augmented reality systems, the two-dimensional (2D) image is used to provide orientation information [8–12]. In a more complex environment, Ref. [13] proposed a method to identify the complete 3D curvature of a flexible marker using five cameras. The system did capture the 3D information needed, but the camera calibration process was too complex. On the other hand, Ref. [1] reformed the 3D structure of the object using a mono-imaging device on a robot arm using the silhouette method. However, the 3D structure reformation process involves multiple angular movements and consumes more computation time. Furthermore, the augmented reality concepts can be exploited to support object-grasping robots [2,3,11]. For example, Ref. [2] built a grasping system using artificial markers and a stereo vision system. This simplifies the target detection process and ensures the system's robustness. However, stereo camera positioning systems involve costly baseline calibration processes [3].

Stereo vision systems can be simplified to mono-imaging systems with artificial landmarks. A previous study [2] used customized markers, called VITags, in their automated picking warehouse, in which the VITag technology is not available to the public, and its functionalities are limited. In addition, Ref. [14] measured the distance of a circle marker from the camera using a mono-imaging system. However, the circular marker could not provide the marker's orientation. The existing 2D barcode technologies can be used as artificial landmarks that provide location and orientation information [6,7,15,16]. For example, Ref. [15] developed a fully automated robotic library system with the aid of a stereo camera system. The system uses QR code landmarks and book labels to ease the positioning process. QR codes are now widely applied due to their high producibility at low cost. Moreover, they can be designed with various sizes, and are detectable even when partially damaged [17]. However, QR codes in most L&N system serve as fixed references and provide only 2D positions. A previous study [18] utilized a QR code's corner point information to identify the distortion level, but the distance of the QR code from the camera was still unknown. Object-grasping automation tasks require position and orientation information in the 3D space. Thus, the current system has limitations in 3D positioning using the information obtained from the QR codes, specifically for monocular vision systems. There is very limited research performed on 3D positioning using a monocular camera and QR code. To the best of our knowledge, Ref. [19] is the only scholar who has implemented a monocular vision-based 3D positioning system with two parallel QR code landmarks. Although the system functions well, the use of two QR codes placed side-by-side as artificial landmarks on existing grocery product packaging is challenging.

The use of more lightweight positioning systems is essential for developing automated pick-cart robots. The current vision positioning systems require two or more optical sensors or multiple landmarks. Some automation systems perform positioning with a single camera but need to capture a few photos from distinct optical points. The process is complicated, from identifying the camera distortion parameters, performing image rectification and matching, followed by positioning. This paper proposes a method for positioning using a mono-imaging system's image captured from one optical point. It simplifies the intricate camera calibration process and uses a pre-processing analysis for multiple images. However, the use of the computation method to yield the target's location information, such as the position, depth, size, and orientation from a single image using one QR code marker, remains an open issue to be solved. Therefore, developing a new approach to reduce the complexity and heavy computation costs of the mono-imaging-based positioning system using numerical computation is the main focus of this paper.

The remainder of this paper is structured as follows. Section 2 presents the QR code's geometric primitives. Section 3 describes the mathematical model that defines the position of the QR code. Then, Section 4 explains the distance model that is used to formulate the updating rules for the positioning model. Section 5 shows the conditioning rules using the QR marker's side length to update the guess value in the numerical method to obtain the

position information. Additionally, a detailed numerical computation of the object point from the image point is provided. Section 6 describes the calculation required to obtain the orientation of the marker. In addition, Section 7 compares the proposed system with the previous work. Next, Section 8 describes the parameters and conditions for the simulation while Section 9 presents the results and discussion. Finally, Section 10 concludes this paper with significant and possible future research directions.

## 2. The 2D QR Code Marker

The QR code has been extensively adopted for product identification recently. It will become a standard marker on every product packaging in the near future. The four corners of the QR code on the packaging can be utilized to identify the position and orientation of the product. The product's position and orientation, together with the packaging's dimension information, can then be used by the robot to compute the gripper trajectory and grasp the product. In line with the current trend, QR codes are used as markers for a product position and orientation identification system in this project. Figure 1 shows the QR code samples with labelled corner points and lines. These points can be used to indicate the QR code's orientation [19,20]. By mounting a camera on the robot end-effector, the QR code image can be acquired. The corners of the QR code can be identified later using an image processing algorithm. The details of the algorithm can be found in [16,21,22]. In this project, the corners are labelled and denoted as $P_1 = (x_1, y_1, z_1)$, $P_2 = (x_2, y_2, z_2)$, $P_3 = (x_3, y_3, z_3)$, and $P_4 = (x_4, y_4, z_4)$, respectively, as shown in Figure 1. Meanwhile, Table 1 lists the details of the length, $l_1$ to $l_6$, between the corresponding points.

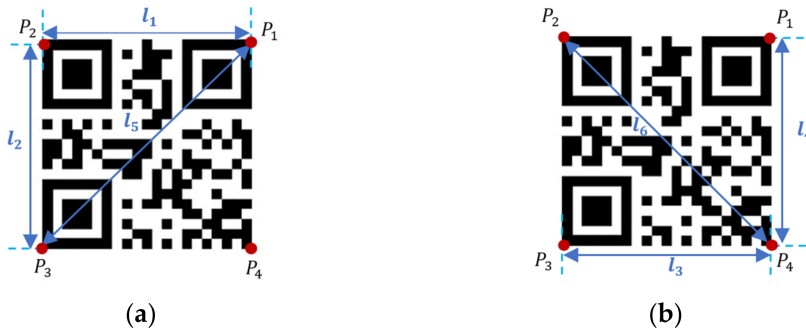

(**a**)          (**b**)

**Figure 1.** Sample QR code: (**a**) length $l_1, l_2, l_5$; (**b**) length $l_3, l_4, l_6$.

**Table 1.** The length of the QR code marker between corresponding points.

| Length ($P_i, P_j$) | Details | Point, $P_i$ | Point, $P_j$ |
|---|---|---|---|
| $l_1$ | Horizontal | $P_1$ | $P_2$ |
| $l_2$ | Vertical | $P_2$ | $P_3$ |
| $l_3$ | Horizontal | $P_3$ | $P_4$ |
| $l_4$ | Vertical | $P_1$ | $P_4$ |
| $l_5$ | Diagonal | $P_1$ | $P_3$ |
| $l_6$ | Diagonal | $P_2$ | $P_4$ |

## 3. The Positioning Model of the QR Code

Figure 2 shows the main workflow used to develop the mathematical model for the proposed positioning system:

1. The system extracts the QR code's 2D image coordinates and maps them to their corresponding 3D geometrical characteristics;
2. The estimated distance between two corner points scales the estimated conditions of the z coordinates. Note that the QR code's z coordinates are the guessing parameters for the numerical computation;
3. The z coordinate values for the next iteration are computed using the updating rules derived based on the difference between the calculated and actual QR code length;
4. The result converges when the absolute convergence error fulfils the requirement;

5.     The orientation information is calculated using the inverse rotation matrix.

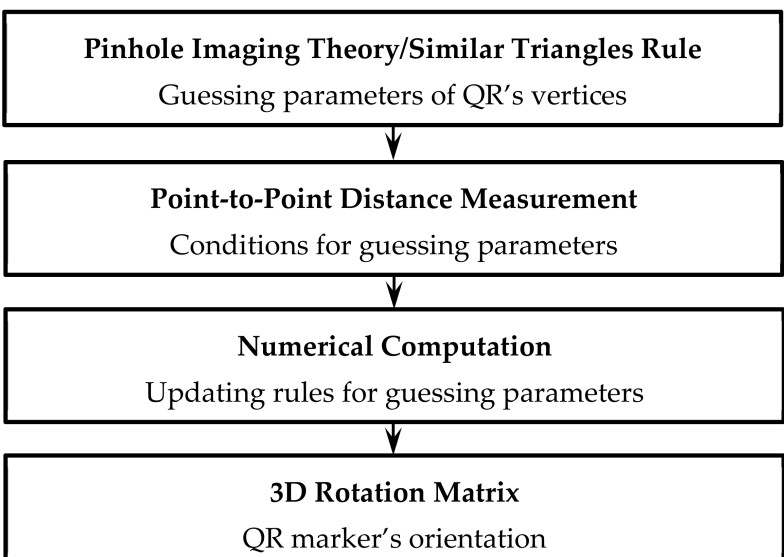

**Figure 2.** The main workflow for the mathematical model.

Decoding the information stored in the QR code is not the main aim of this research.

The pinhole imaging theory and similar triangles rule are adopted to determine the object's location from an image [12,14,23]. This relates the object to the corresponding 2D image, as illustrated in Figure 3, where $L$ denotes the actual side length of the object, $l$ is the side length of the object in the captured image, d represents the working distance between the object and the camera lens, and $f$ is the focal length or the distance between the lens and the camera sensor.

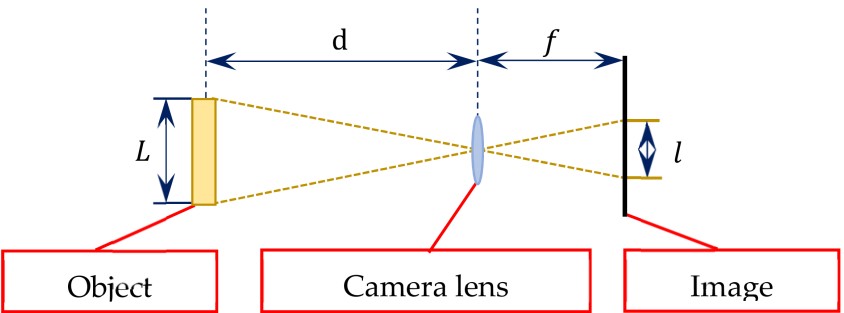

**Figure 3.** The basic principle of pinhole imaging.

Referring to pinhole imaging theory, the ratio of the focal length to the object distance can be defined as:

$$\frac{f}{d} = \frac{l}{L} \tag{1}$$

Thus, the actual side length of the object is:

$$L = \frac{l}{f}d \tag{2}$$

Figure 4 illustrates the forward perspective projection in a 3D environment. The camera lens is referred to as a zero-reference coordinate frame, while the focus point is reflected towards the positive z direction. Referring to the figure, $P = (x, y, z)$ is the object point that corresponds to the image points $P' = (x', y', z')$.

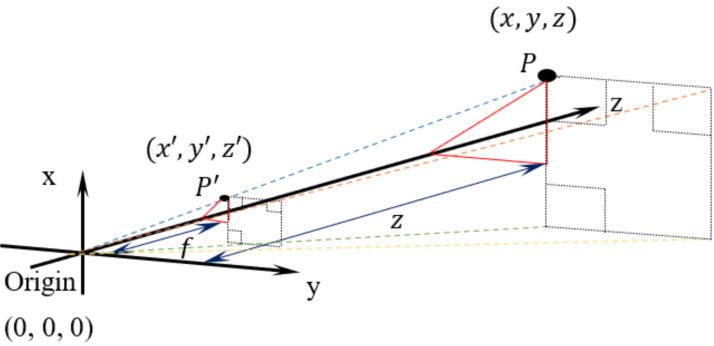

**Figure 4.** The 3D coordinate estimation of the marker point.

Figure 4 shows the camera's focal length, $f$, which is also the $z'$ coordinate of the image point $P'$. Thus, the image point can also be defined as $P' = (x', y', f)$. Then, using the pinhole imaging theory and similar triangles rule in Equation (2), the relationship between the real-world point and image point is:

$$\frac{f}{z} = \frac{P'}{P} \tag{3}$$

The x-coordinate of object point $P$ is:

$$\begin{aligned} \frac{f}{z} &= \frac{x'}{x} \\ x &= \frac{x'}{f}(z) \end{aligned} \tag{4}$$

The y-coordinate of object point $P$ is:

$$\begin{aligned} \frac{f}{z} &= \frac{y'}{y} \\ y &= \frac{y'}{f}(z) \end{aligned} \tag{5}$$

Substituting the QR code's vertices, $P_1$ to $P_4$, as the object point $P$, the QR code's x-coordinate and y-coordinate can be defined in a general form as:

$$\begin{cases} x_i = \frac{x_i'}{f}(z_i) \\ y_i = \frac{y_i'}{f}(z_i) \end{cases}, \ i = 1, 2, 3, 4 \tag{6}$$

where $x_i'$ and $y_i'$ represent the QR code image points' x- and y-coordinates, respectively, and $x_i$, $y_i$, and $z_i$ respectively represent the x-, y- and z-coordinates of the QR code's vertices in a real-world environment, concerning the zero-reference frame.

Referring to Equation (6), the 2D image points' coordinates and the real-world z-coordinates can be used to compute the x- and y-coordinates of the QR code. The 2D image coordinate information can be extracted from the captured image, while the z-coordinate is an unknown parameter. Thus, the z-coordinate is set as the guessing parameter in the numerical computation. In addition, proper updating rules are required to update the guessed z-coordinate value after each iteration in the numerical computation.

## 4. Point-to-Point Distance Model as Z-Coordinate Conditioning Rules

The guessed conditions for the z-coordinates require a known value to gauge the estimated value. Since the distance between the corner points $P_i$ and $P_j$ on the QR code is known and fixed, it can be used to formulate the updating rules. Referring to Table 1 and using the Pythagorean theorem [24], the distance between the corner points, $l_{i,j}$, is defined as:

$$l_{i,j} = \sqrt{(x_i - x_j)^2 + (y_i - y_j)^2 + (z_i - z_j)^2}, \ for \ i \neq j, \ i = 1, 2, 3, 4 \tag{7}$$

The QR code is square-shaped; thus, its horizontal and vertical length are the same.

$$l_1 = l_2 = l_3 = l_4 = l \tag{8}$$

Meanwhile, using trigonometry rules [24,25], as shown in Figure 5, the diagonal distance $l_5$, $l_6$ can also be defined as:

$$l_5 = l_6 = 2l \cos \theta \; ; \; \theta = 45° \tag{9}$$

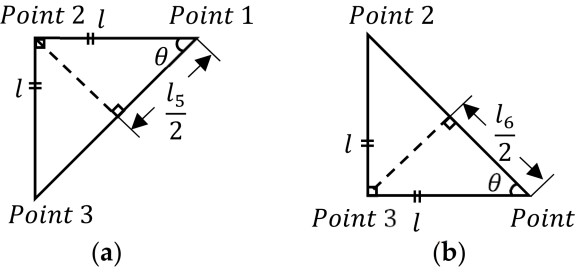

**Figure 5.** Trigonometry rules for length calculation: (**a**) length $l_5$; (**b**) length $l_6$.

## 5. QR's Points Positioning Using Numerical Method

This paper proposes a lightweight numerical method to overcome the complexity of the QR corner point computation.

### 5.1. Guessed Z-Value Conditions

The estimation condition for the z-coordinate can be identified using the QR's side length or diagonal length calculated from Equation (7). Then, the condition error, $e_i$ $(for \; i = 1, 2, 3, 4, 5, 6)$, as defined in Equations (10) and (11), is used to formulate the updating rules of the z-coordinate for the next iteration.

The conditions when using the horizontal or vertical side length, $l_1$, $l_2$, $l_3$, $l_4$, are:

1. If $\sqrt{(x_i - x_j)^2 + (y_i - y_j)^2 + (z_i - z_j)^2} > l$, either $z_i$ or $z_j$ or both $z_i$ and $z_j$ are too big;
2. If $\sqrt{(x_i - x_j)^2 + (y_i - y_j)^2 + (z_i - z_j)^2} < l$, either $z_i$ or $z_j$ or both $z_i$ and $z_j$ are too small;
3. If $\sqrt{(x_i - x_j)^2 + (y_i - y_j)^2 + (z_i - z_j)^2} = l$, then $z_i$ and $z_j$ is the right guess.
4. The condition error related to the horizontal and vertical side length is:

$$e_i = l_i - \sqrt{(x_i - x_j)^2 + (y_i - y_j)^2 + (z_i - z_j)^2} \; , \; \text{for } i = 1, 2, 3, 4 \tag{10}$$

The conditions when using diagonal length, $l_5$, $l_6$, are:

1. If $\sqrt{(x_i - x_j)^2 + (y_i - y_j)^2 + (z_i - z_j)^2} > 2l cos45°$, either $z_i$ or $z_j$ or both $z_i$ and $z_j$ are too big;
2. If $\sqrt{(x_i - x_j)^2 + (y_i - y_j)^2 + (z_i - z_j)^2} < 2l cos45°$, either $z_i$ or $z_j$ or both $z_i$ and $z_j$ are too small;
3. If $\sqrt{(x_i - x_j)^2 + (y_i - y_j)^2 + (z_i - z_j)^2} = 2l cos45°$; $z_i$ and $z_j$ hits the right guess.

The condition error related to the diagonal length is:

$$e_i = 2l cos45° - \sqrt{(x_i - x_j)^2 + (y_i - y_j)^2 + (z_i - z_j)^2} \; , \; \text{for } i = 5, 6 \tag{11}$$

Referring to Figure 1 and Table 1, three corresponding lengths, $l_1$, $l_4$, and $l_5$, are used to justify the conditions when updating the value of $z_1$. Similarly, one can justify the value of $z_2$, $z_3$, and $z_4$ in the same way. Thus, the updating rule for each value of $z_i$, $i = 1, 2, 3, 4$ corresponds with three condition errors is tabulated in Table 2.

**Table 2.** The combination of the condition error, $e_i$ with the corresponding $z_i$.

| z-Coordinate | $e_1$ | $e_2$ | $e_3$ | $e_4$ | $e_5$ | $e_6$ |
|:---:|:---:|:---:|:---:|:---:|:---:|:---:|
| $z_1$ | / | | | / | / | |
| $z_2$ | / | / | | | | / |
| $z_3$ | | / | / | | / | |
| $z_4$ | | | / | / | | / |

### 5.2. Updating Rules

The updating rules for the subsequent guessed value, $z_i(n+1)$, with $k$ as the updating coefficient and n as the number of iterations, are defined as:

$$z_i(n+1) = z_i(n) + \frac{k(e_{j,1} + e_{j,2} + e_{j,3})}{3} \text{, for } i = 1,2,3,4 \text{ ; } j = 1,2,3,4,5,6 \qquad (12)$$

The z-coordinate value for the corner points can be updated using the old value plus the average error, as per Equation (12), based on the corresponding condition error listed in Table 2 at each iteration until the numerical computation converges to a stable value within the acceptable convergence error range. The sum of the absolute convergence error, $e_c$, is defined as the absolute sum of the current $z_i(n)$ minus the previous $z_i(n-1)$ as per Equation (13). The desired sum of the absolute convergence error, $e_d$, is set as the acceptable error, where convergence is achieved when $e_c \leq e_d$.

$$e_c = \sum_{i=1}^{4} |z_i(n) - z_i(n-1)| \qquad (13)$$

### 5.3. Numerical Computation

The flow of the numerical computation is shown in Figure 6. During the numerical computation, the values of $x_i$ and $y_i$ for $i$ = 1, 2, 3, 4 are computed using Equation (6), based on the assumed $z_i$, given $f$, and image point coordinates. An initial guess value of $z_{initial}$ is used at the beginning of the computation and the $z_i\ for\ i = 1,2,3,4$ is then updated at each iteration using Equation (12) with reference to Table 2. Next, substituting the previous and updated $z_i$ values into Equation (13) will yield the sum of the absolute convergence error, $e_c$, which is used for comparison with the desired sum of the absolute convergence error, $e_d$. In the meantime, to avoid an infinite looping state, the maximum number of iterations, $n_{max}$, is set. The computation will exit the loop when $e_c \leq e_d$ or $n > n_{max}$ is achieved.

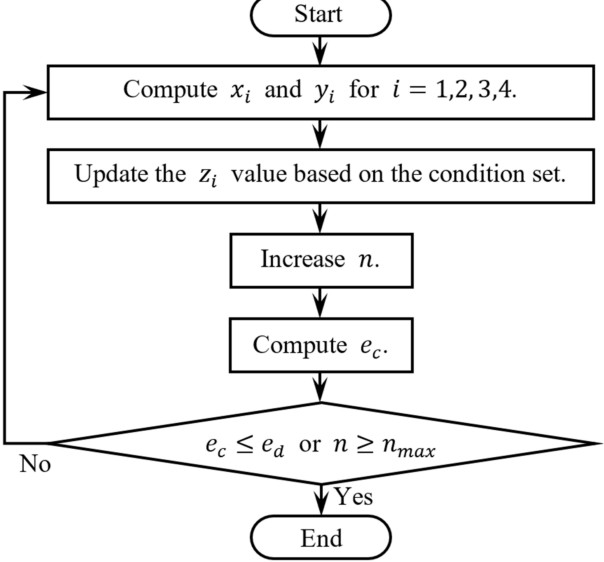

**Figure 6.** Flowchart of the numerical computation.

## 6. Orientation Calculation of 2D QR Code Marker

The orientation of the QR code marker can be obtained from a 3D rotation matrix using the three basic counterclockwise rotation vectors about the x-, y-, and z-axes [26], as shown below.

$$R_x(\alpha) = \begin{pmatrix} 1 & 0 & 0 \\ 0 & \cos\alpha & -\sin\alpha \\ 0 & \sin\alpha & \cos\alpha \end{pmatrix} \tag{14}$$

$$R_y(\beta) = \begin{pmatrix} \cos\beta & 0 & \sin\beta \\ 0 & 1 & 0 \\ -\sin\beta & 0 & \cos\beta \end{pmatrix} \tag{15}$$

$$R_z(\gamma) = \begin{pmatrix} \cos\gamma & -\sin\gamma & 0 \\ \sin\gamma & \cos\gamma & 0 \\ 0 & 0 & 1 \end{pmatrix} \tag{16}$$

The 3D rotation matrix for the $R_x(\alpha)R_y(\beta)R_z(\gamma)$ system, which rotates angle $\alpha$ about the x-axis, followed by angle $\beta$ about the y-axis and then angle $\gamma$ about the z-axis, is shown in Equation (17):

$$R(\alpha, \beta, \gamma) = \begin{pmatrix} C_\beta C_\gamma & -C_\beta S_\gamma & S_\beta \\ S_\alpha S_\beta C_\gamma + C_\alpha S_\gamma & C_\alpha C_\gamma - S_\alpha S_\beta S_\gamma & -S_\alpha C_\beta \\ S_\alpha S_\gamma - C_\alpha S_\beta C_\gamma & C_\alpha S_\beta S_\gamma + S_\alpha C_\gamma & C_\alpha C_\beta \end{pmatrix} \tag{17}$$

where $S_{(\theta)}$ and $C_{(\theta)}$ denote $\sin\theta$ and $\cos\theta$ $for\ \theta = \alpha, \beta, \gamma$ respectively. The local coordinate system parallel to the gripper coordinate using the QR marker point $P_2$, shown in Figure 1, as the base point of the rotation is defined as $\hat{P}_i\ for\ i = 1, 2, 3, 4$. Thus, at $\alpha = \beta = \gamma = 0$, the QR vertices with reference to the origin $\hat{P}_{o,2} = (0,\ 0,\ 0)$ are,

$$\begin{aligned} \hat{P}_{o,1} &= (0,\ l,\ 0) \\ \hat{P}_{o,3} &= (-l,\ 0,\ 0) \\ \hat{P}_{o,4} &= (-l,\ l,\ 0) \end{aligned} \tag{18}$$

On the other hand, the result from the numerical computation provides the coordinates of QR code's corner points with reference to the coordinate system based on the camera mounted on the gripper. For this situation, Equation (19) is used to transform the coordinates computed with the local rotation-based point coordinate system $\hat{P}_i$.

$$\hat{P}_i\ (\hat{x}_i,\ \hat{y}_i,\ \hat{z}_i) = (x_i - x_2,\ y_i - y_2,\ z_i - z_2),\ for\ i = 1, 3, 4 \tag{19}$$

Referring to Equations (17) and (18), $\hat{P}_i$ also can also be yielded as a result of the rotation of point $\hat{P}_{oi}$ as:

$$\hat{P}_1 = R(\alpha, \beta, \gamma)\hat{P}_{o,1}{}^T$$

$$= \begin{pmatrix} l(-C_\beta S_\gamma) \\ l(C_\alpha C_\gamma - S_\alpha S_\beta S_\gamma) \\ l(C_\alpha S_\beta S_\gamma + S_\alpha C_\gamma) \end{pmatrix} \tag{20}$$

$$\hat{P}_3 = R(\alpha, \beta, \gamma)\hat{P}_{o,3}{}^T$$

$$= \begin{pmatrix} -l(C_\gamma C_\beta) \\ -l(S_\alpha S_\beta C_\gamma + C_\alpha S_\gamma) \\ -l(S_\alpha S_\gamma - C_\alpha S_\beta C_\gamma) \end{pmatrix} \tag{21}$$

From $\hat{x}_1$ and $\hat{x}_3$ in Equations (19)–(21), the value of angle $\gamma$ that rotates about the z-axis can be obtained.

$$\gamma = \tan^{-1}\left[\frac{\hat{x}_1(l_2)}{\hat{x}_3(l_1)}\right] \tag{22}$$

The value of $\gamma$ is then substituted into $\hat{x}_1$ from Equation (20) to get the value of $\beta$, the angle of rotation about the y-axis.

$$\beta = -\cos^{-1}\left[\frac{\hat{x}_1}{\sin\gamma(l_1)}\right] \tag{23}$$

Lastly, the value of angle $\alpha$ that rotates about the x-axis can be obtained after substituting $\gamma$ and $\beta$ into $\hat{y}_1$ and $\hat{z}_1$ from Equations (20).

$$\alpha = \cos^{-1}\left[\frac{\sin\beta\tan\gamma(\hat{z}_1) + \hat{y}_1}{l_1(\cos\gamma + \sin\beta\sin\beta\tan\gamma\sin\gamma)}\right] \tag{24}$$

## 7. Comments on the Proposed Method and Comparison with Previous Work

A previous study [19] proposed a 3D positioning model using a monocular camera and two QR codes. The two QR code landmarks are placed side-by-side to improve the detection accuracy. The previous study [19] used six positioning points, while this paper only uses four points. This paper uses only one QR code marker for the real-life application, as only one QR code will be printed on most grocery packaging. As the system proposed by [19] requires two QR codes for positioning purposes, so it is not practical to identify the position of the grocery goods using the single QR code on the grocery packaging. On the other hand, although the QR code on the grocery packaging come in various sizes, the QR code's size and information can be registered and updated easily through the current proposed system. Additionally, this paper uses a 50 mm QR code, which is smaller than the 120 mm QR codes used by [19].

The proposed method has advantages over the other monocular-based positioning systems (to the best of our knowledge):

- The numerical computation solves the underdetermined positioning system with simple arithmetic operations, using a lightweight positioning method with four positioning points;
- It works with basic geometry principles and trigonometry relations, and results in a minor error of floating points;
- It can perform position and orientation estimations with the known camera's focal length and QR code's size, meaning the camera calibration process can be diminished;
- It can extract the depth information between the QR code landmark and the monocular camera using one QR code image captured at a fixed optical point; rectification and matching of the images are unnecessary.

On the other hand, the previous study [19] uses an efficient perspective-in-point (EPnP) algorithm for camera pose calculation. The system is able to calculate the 3D position at all locations. However, the proposed numerical computation method in this paper might provide diverged results, which can mean the positional information is not computed at a certain location. Additionally, the translation parameters such as the z working distance and x translation used in [19] are higher than the parameters set in this paper, which is due to the size of the QR code landmark used in [19] being 1.4 time larger than the one used in this paper.

## 8. Positioning Model Simulation Using MATLAB

MATLAB is the numerical computing platform used to simulate the proposed positioning model. The simulation was configured in MATLAB version R2017b and performed on a Windows 10 desktop with an Intel CPU Core i7-3517U with 1.90 GHz and 8 GB RAM. The authors fully implemented the MATALB code and no library was used. Figure 7 shows the main workflow of the simulation model. First, the theoretical image coordinates of the QR code's vertices are estimated based on the rotation matrix and the QR code's world coordinates. Then, the proposed positioning model computes the 3D position information using Equation (6). The initial value of 450 mm is set for the guessing parameters,

z-coordinate of the QR code's corner points. Then, it will update for the next iteration based on the updating rules as per Equation (12). Furthermore, the convergence error, $e_c$, scales the estimating conditions of the guessing parameters. The simulation is looped until the maximum number of loops, $n_{max}$, is reached or when the absolute convergence error is within the tolerance range.

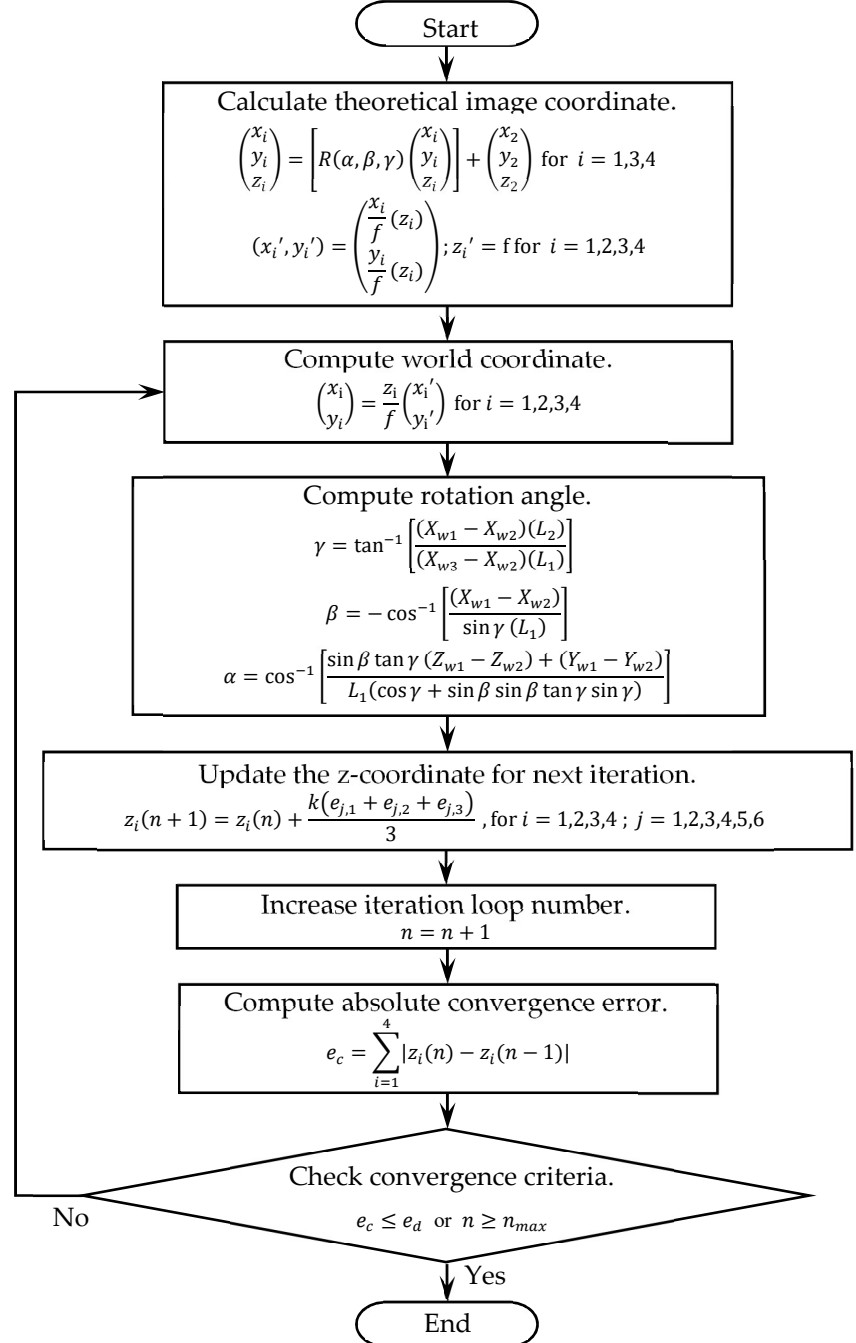

**Figure 7.** Main workflow of the MATLAB numerical simulation model.

Simulations are conducted to study the maximum value of the rotation angle achievable with different rotation combination sets based on the parameters listed in Table 3. The QR code's size and the camera's focal length are the only given parameters for the positioning model. The length of the QR code is set as 50 mm and the camera's focal length is set as 2.8 mm. In addition, the value of the rotation angle about the cardinal axes and notations for the 12 rotation combination sets are listed in Table 4.

**Table 3.** Parameters for the MATLAB simulation.

| Parameters | Value |
|---|---|
| Size of QR marker, $L_1 \times L_2$ | 50 mm $\times$ 50 mm |
| Camera's focal length | 2.82 mm |

**Table 4.** Rotation combination sets for the simulation.

| Combination Sets | Value of Rotation Angle around the Axis | | |
|---|---|---|---|
| | x, Angle $\alpha$ (°) | y, Angle $\beta$ (°) | z, Angle $\gamma$ (°) |
| X—0—0 | $x_{range}$ | 0 | 0 |
| X—0—5 | $x_{range}$ | 0 | 5 |
| X—5—0 | $x_{range}$ | 5 | 0 |
| X—5—5 | $x_{range}$ | 5 | 5 |
| 0—Y—0 | 0 | $y_{range}$ | 0 |
| 0—Y—5 | 0 | $y_{range}$ | 5 |
| 5—Y—0 | 5 | $y_{range}$ | 0 |
| 5—Y—5 | 5 | $y_{range}$ | 5 |
| 0—0—Z | 0 | 0 | $z_{range}$ |
| 0—5—Z | 0 | 5 | $z_{range}$ |
| 5—0—Z | 5 | 0 | $z_{range}$ |
| 5—5—Z | 5 | 5 | $z_{range}$ |

The simulation is repeated for each rotation combination set, and the maximum angle of rotation at each coordinate point is determined. For each simulation, $n_{max}$ is 30 loops and the $e_d$ value of 1 is set. Additionally, the error of the calculation for the rotation angle $(\alpha, \beta, \gamma)$ is not more than 2° and the error of distance ($z_i$, for $i = 1, 2, 3, 4$) is within 5 mm. Using the initial assumption value, $z_i = 450$, the simulation is conducted for the range:

1. X-axis translation from $-100$ mm to $+100$ mm, with a 10 mm step size;
2. Y-axis translation from $-100$ mm to $+100$ mm, with a 10 mm step size;
3. Z-axis translation from $+100$ mm to $+500$ mm, with a 100 mm step size;
4. Angle of rotation ($x_{range}, y_{range}, z_{range}$) from 0° to 25°, with a 1° increment;
5. Updating coefficient $k$ from 0.8 to 2.0, with an increment of 0.1.

## 9. Results and Discussions

Figures 8–12 illustrate the simulation results for the positioning model at the working distance from 500 mm to 100 mm. The coordinate of the center of rotation, *point* 2 of the QR marker, is used as the reference point to plot the results. At the given location $(x, y)$ with the working distance z, the maximum rotation angle for the respective combination sets is tabulated graphically on the coordinate plane. The coordinate plane's horizontal axis represents the y-axis translation. On the vertical axis is the translation in the x-axis direction. Moreover, the working distance is the translation in the z-axis direction of *point* 2 away from the camera origin. The color gradient indicates the solvable value of the angle of rotation, with yellow representing the maximum value of 25° and dark blue representing the diverged result.

The simulation results for the maximum angle of rotation achievable at each coordinate at a working distance of 500 mm are as shown in Figure 8.

The simulation results for the maximum angle of rotation achievable at each coordinate at a working distance of 400 mm are as shown in Figure 9.

The simulation results for the maximum angle of rotation achievable at each coordinate at a working distance of 300 mm are as shown in Figure 10.

The simulation results for the maximum angle of rotation achievable at each coordinate at a working distance of 200 mm are as shown in Figure 11.

The simulation results for the maximum angle of rotation achievable at each coordinate at a working distance of 100 mm are as shown in Figure 12.

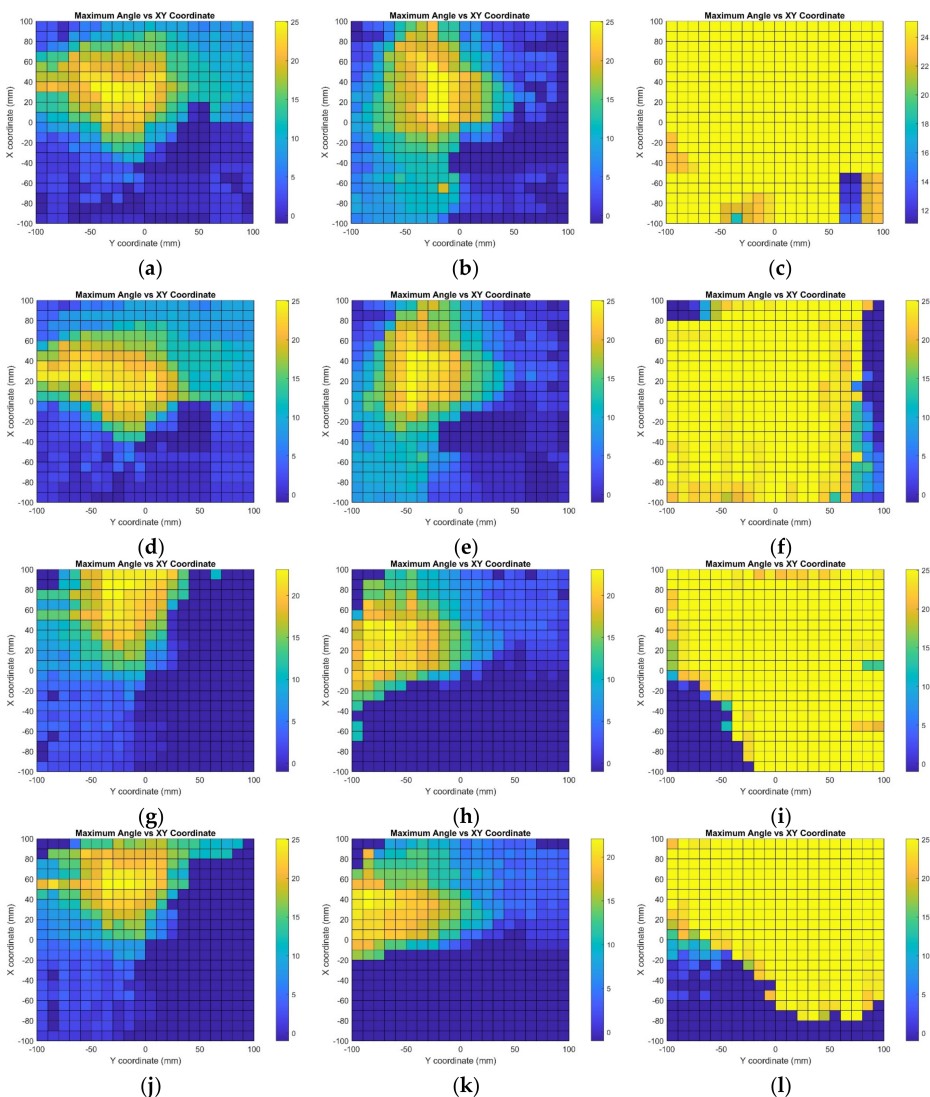

**Figure 8.** Simulation results for the working distance of 500 mm for rotation combination sets: (**a**) X—0—0; (**b**) 0—Y—0; (**c**) 0—0—Z; (**d**) X—0—5; (**e**) 0—Y—5; (**f**) 0—5—Z; (**g**) X—5—0; (**h**) 5—Y—0; (**i**) 5—0—Z; (**j**) X—5—5; (**k**) 5—Y—5; (**l**) 5—5—Z.

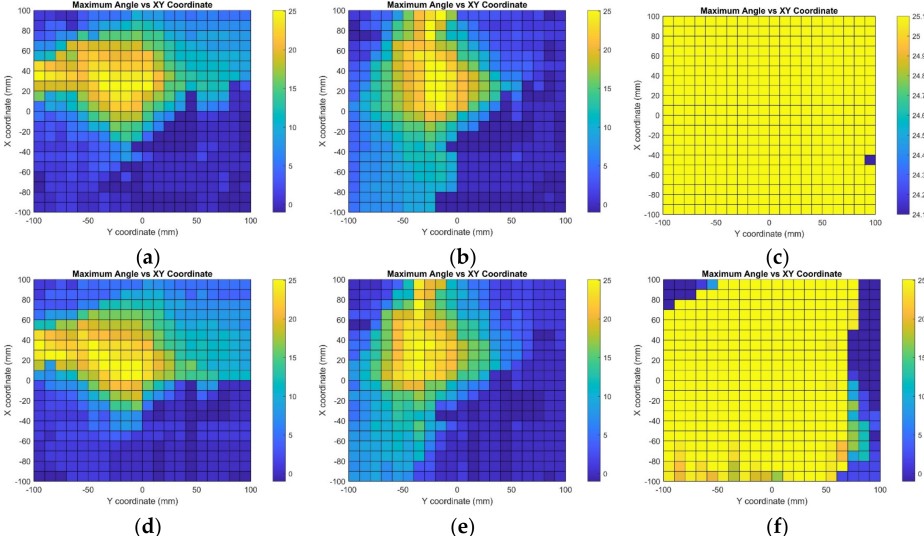

**Figure 9.** *Cont.*

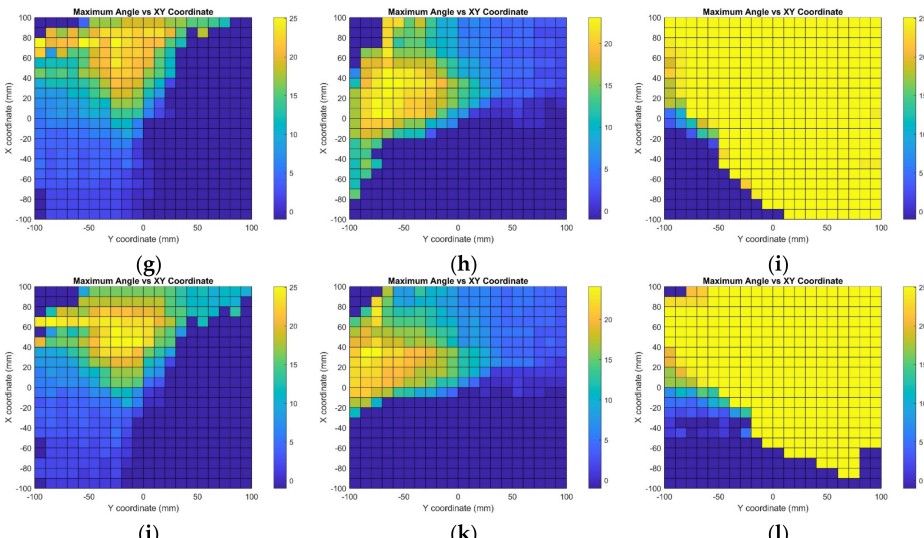

**Figure 9.** Simulation results for the working distance of 400 mm for rotation combination sets: (**a**) X—0—0; (**b**) 0—Y—0; (**c**) 0—0—Z; (**d**) X—0—5; (**e**) 0—Y—5; (**f**) 0—5—Z; (**g**) X—5—0; (**h**) 5—Y—0; (**i**) 5—0—Z; (**j**) X—5—5; (**k**) 5—Y—5; (**l**) 5—5—Z.

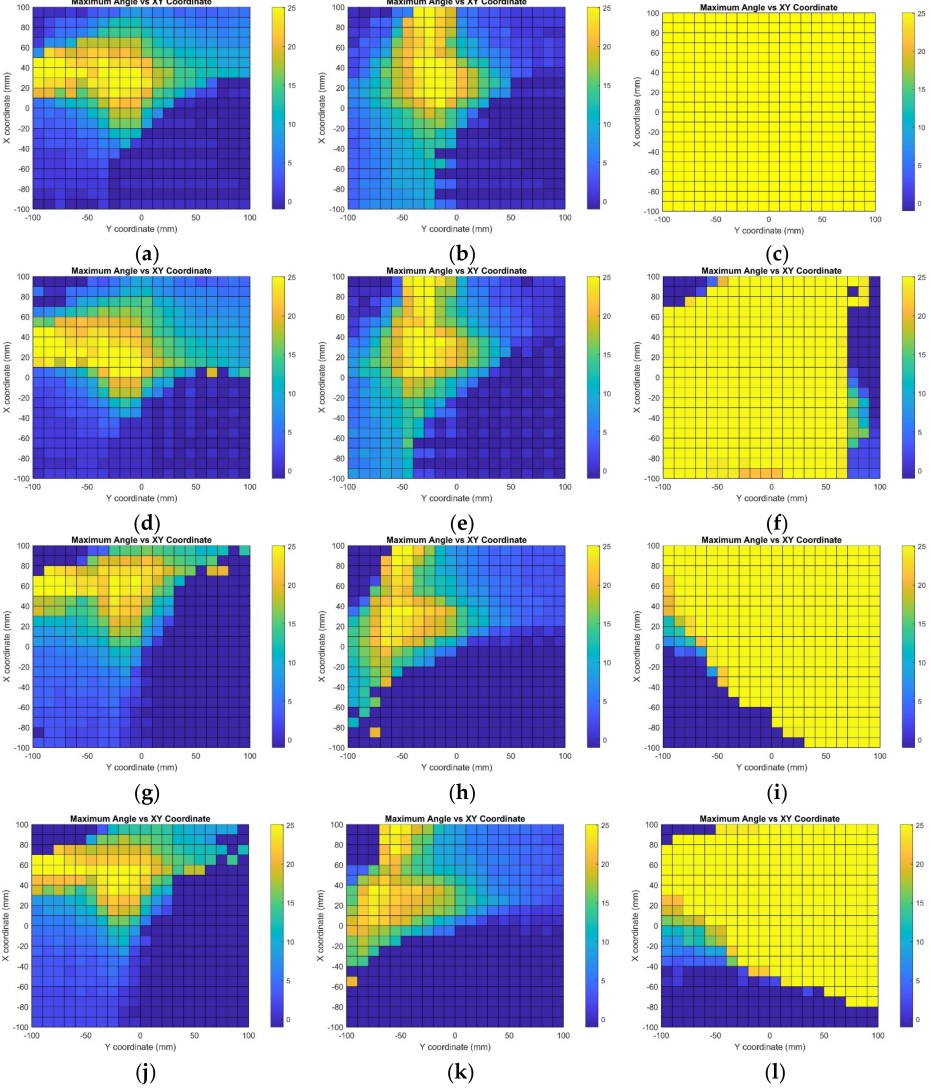

**Figure 10.** Simulation results for the working distance of 300 mm for rotation combination sets:

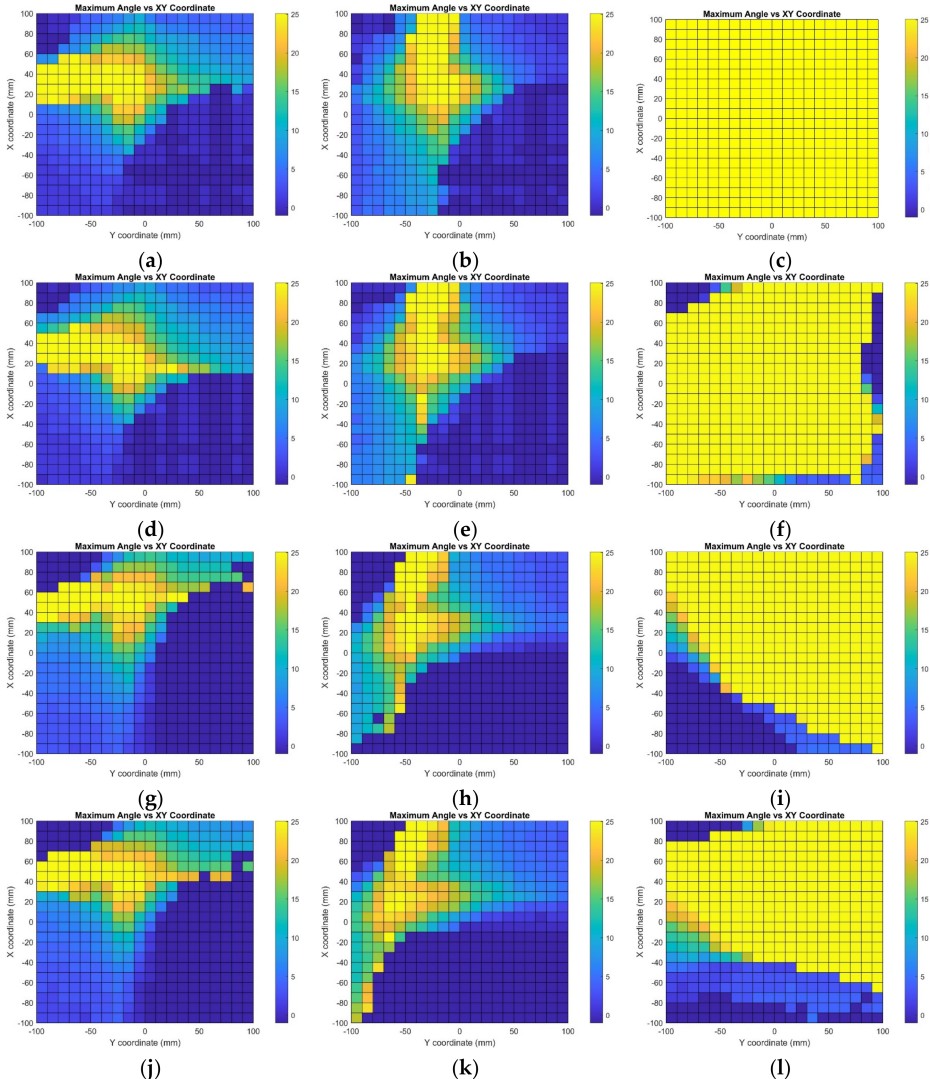

**Figure 11.** Simulation results for the working distance of 200 mm for rotation combination sets: (**a**) X—0—0; (**b**) 0—Y—0; (**c**) 0—0—Z; (**d**) X—0—5; (**e**) 0—Y—5; (**f**) 0—5—Z; (**g**) X—5—0; (**h**) 5—Y—0; (**i**) 5—0—Z; (**j**) X—5—5; (**k**) 5—Y—5; (**l**) 5—5—Z.

### 9.1. Result Validation

This section compares the calculated values of the rotation angle and 3D coordinates with the simulated value to validate the results. A point selected from each rotation combination at five working distances from 500 mm to 100 mm is used for comparison. The simulation results show that the positioning model achieves satisfactory accuracy, and this is verified by the monocular single QR code image positioning and orientation results using the numerical system. The average error achieved is not more than two degrees and there is less than 5 mm difference for the simulated values, as shown in Tables 5–9.

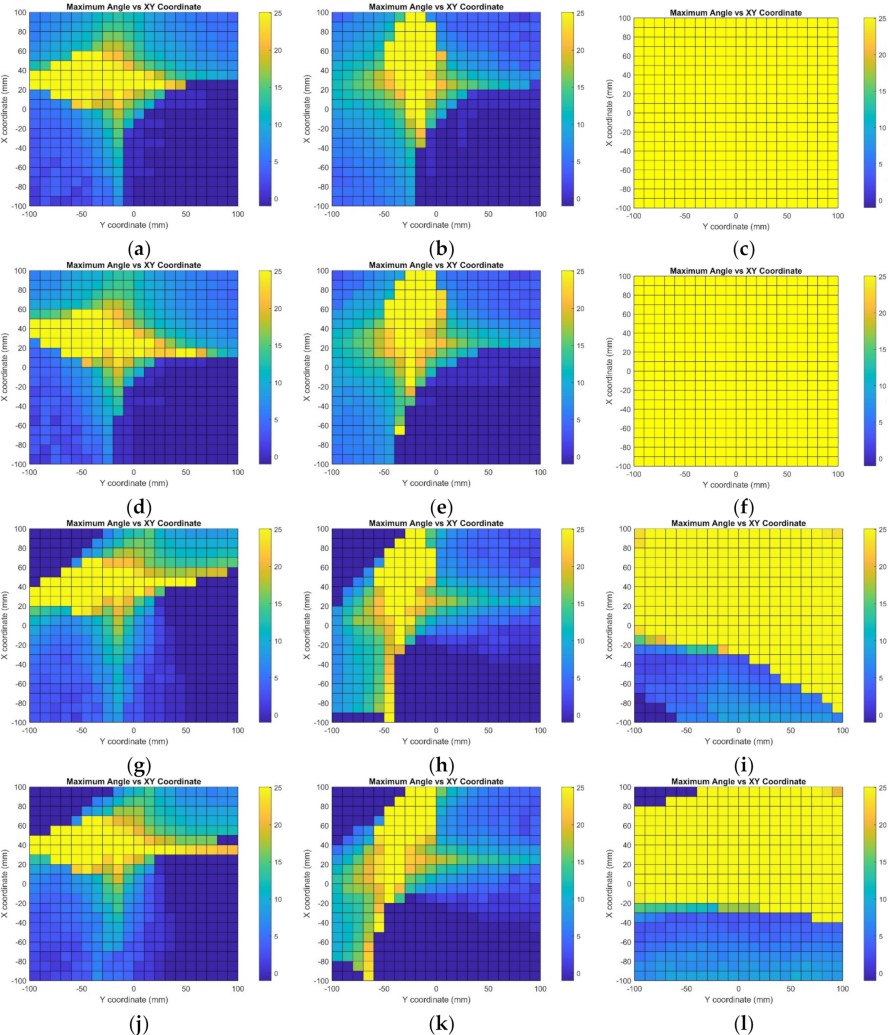

**Figure 12.** Simulation results for the working distance of 100 mm for rotation combination sets:
(**a**) X—0—0; (**b**) 0—Y—0; (**c**) 0—0—Z; (**d**) X—0—5; (**e**) 0—Y—5; (**f**) 0—5—Z; (**g**) X—5—0; (**h**) 5—Y—0;
(**i**) 5—0—Z; (**j**) X—5—5; (**k**) 5—Y—5; (**l**) 5—5—Z.

**Table 5.** Comparison between actual values and computed results at 500 mm.

| Combination Sets | Calculated Value | | | | Simulated Value | | | |
|---|---|---|---|---|---|---|---|---|
| | Angle (°) | Coordinate (mm) | | | Angle (°) | Coordinate (mm) | | |
| | | *x* | *y* | *z* | | *x* | *y* | *z* |
| X—0—0 | 22.00 | 60.00 | −50.00 | 500.00 | 23.22 | 59.18 | −49.48 | 493.97 |
| X—0—5 | 16.00 | 60.00 | −50.00 | 500.00 | 15.69 | 59.54 | −49.78 | 496.98 |
| X—5—0 | 23.00 | 60.00 | −50.00 | 500.00 | 21.16 | 60.40 | −50.49 | 504.10 |
| X—5—5 | 25.00 | 60.00 | −50.00 | 500.00 | 24.06 | 60.27 | −50.39 | 503.02 |
| 0—Y—0 | 25.00 | 60.00 | −50.00 | 500.00 | 25.27 | 59.72 | −49.93 | 498.45 |
| 0—Y—5 | 25.00 | 60.00 | −50.00 | 500.00 | 23.60 | 60.38 | −50.48 | 503.99 |
| 5—Y—0 | 20.00 | 60.00 | −50.00 | 500.00 | 18.40 | 60.33 | −50.44 | 503.58 |
| 5—Y—5 | 14.00 | 60.00 | −50.00 | 500.00 | 12.64 | 60.08 | −50.23 | 501.43 |
| 0—0—Z | 25.00 | 60.00 | −50.00 | 500.00 | 25.12 | 59.51 | −49.75 | 496.70 |
| 0—5—Z | 25.00 | 60.00 | −50.00 | 500.00 | 25.35 | 59.62 | −49.85 | 497.65 |
| 5—0—Z | 25.00 | 60.00 | −50.00 | 500.00 | 25.33 | 59.47 | −49.71 | 496.33 |
| 5—5—Z | 25.00 | 60.00 | −50.00 | 500.00 | 25.30 | 59.61 | −49.84 | 497.54 |

**Table 6.** Comparison between actual values and computed results at 400 mm.

| Combination Sets | Calculated Value | | | | Simulated Value | | | |
|---|---|---|---|---|---|---|---|---|
| | Angle (°) | Coordinate (mm) | | | Angle (°) | Coordinate (mm) | | |
| | | *x* | *y* | *z* | | *x* | *y* | *z* |
| X—0—0 | 19.00 | 60.00 | −50.00 | 400.00 | 19.32 | 59.89 | −49.89 | 399.66 |
| X—0—5 | 15.00 | 60.00 | −50.00 | 400.00 | 13.48 | 60.29 | −50.22 | 402.32 |
| X—5—0 | 25.00 | 60.00 | −50.00 | 400.00 | 23.31 | 60.61 | −50.48 | 404.45 |
| X—5—5 | 25.00 | 60.00 | −50.00 | 400.00 | 23.80 | 60.53 | −50.42 | 403.91 |
| 0—Y—0 | 24.00 | 60.00 | −50.00 | 400.00 | 22.92 | 60.24 | −50.17 | 401.95 |
| 0—Y—5 | 25.00 | 60.00 | −50.00 | 400.00 | 23.34 | 60.48 | −50.37 | 403.54 |
| 5—Y—0 | 18.00 | 60.00 | −50.00 | 400.00 | 16.27 | 60.21 | −50.15 | 401.74 |
| 5—Y—5 | 16.00 | 60.00 | −50.00 | 400.00 | 14.32 | 60.20 | −50.14 | 401.67 |
| 0—0—Z | 25.00 | 60.00 | −50.00 | 400.00 | 25.14 | 59.04 | −49.17 | 393.93 |
| 0—5—Z | 25.00 | 60.00 | −50.00 | 400.00 | 24.99 | 59.46 | −49.53 | 396.79 |
| 5—0—Z | 25.00 | 60.00 | −50.00 | 400.00 | 25.18 | 59.41 | −49.48 | 396.44 |
| 5—5—Z | 25.00 | 60.00 | −50.00 | 400.00 | 25.26 | 59.57 | −49.61 | 397.48 |

**Table 7.** Comparison between actual values and computed results at 300 mm.

| Combination Sets | Calculated Value | | | | Simulated Value | | | |
|---|---|---|---|---|---|---|---|---|
| | Angle (°) | Coordinate (mm) | | | Angle (°) | Coordinate (mm) | | |
| | | *x* | *y* | *z* | | *x* | *y* | *z* |
| X—0—0 | 18.00 | 60.00 | −50.00 | 300.00 | 17.71 | 59.96 | −50.03 | 299.90 |
| X—0—5 | 15.00 | 60.00 | −50.00 | 300.00 | 14.05 | 60.03 | −50.08 | 300.21 |
| X—5—0 | 23.00 | 60.00 | −50.00 | 300.00 | 21.15 | 60.48 | −50.46 | 302.50 |
| X—5—5 | 25.00 | 60.00 | −50.00 | 300.00 | 23.23 | 60.61 | −50.57 | 303.13 |
| 0—Y—0 | 23.00 | 60.00 | −50.00 | 300.00 | 21.74 | 60.29 | −50.30 | 301.55 |
| 0—Y—5 | 25.00 | 60.00 | −50.00 | 300.00 | 23.41 | 60.41 | −50.40 | 302.12 |
| 5—Y—0 | 22.00 | 60.00 | −50.00 | 300.00 | 20.12 | 60.56 | −50.53 | 302.89 |
| 5—Y—5 | 18.00 | 60.00 | −50.00 | 300.00 | 16.19 | 60.16 | −50.19 | 300.88 |
| 0—0—Z | 25.00 | 60.00 | −50.00 | 300.00 | 25.15 | 58.92 | −49.16 | 294.70 |
| 0—5—Z | 25.00 | 60.00 | −50.00 | 300.00 | 24.98 | 59.44 | −49.59 | 297.29 |
| 5—0—Z | 25.00 | 60.00 | −50.00 | 300.00 | 25.38 | 58.95 | −49.18 | 294.83 |
| 5—5—Z | 25.00 | 60.00 | −50.00 | 300.00 | 25.29 | 59.46 | −49.61 | 297.38 |

**Table 8.** Comparison between actual values and computed results at 200 mm.

| Combination Sets | Calculated Value | | | | Simulated Value | | | |
|---|---|---|---|---|---|---|---|---|
| | Angle (°) | Coordinate (mm) | | | Angle (°) | Coordinate (mm) | | |
| | | *x* | *y* | *z* | | *x* | *y* | *z* |
| X—0—0 | 14.00 | 60.00 | −50.00 | 200.00 | 12.49 | 60.30 | −50.24 | 200.93 |
| X—0—5 | 15.00 | 60.00 | −50.00 | 200.00 | 13.41 | 60.31 | −50.24 | 200.95 |
| X—5—0 | 24.00 | 60.00 | −50.00 | 200.00 | 22.30 | 60.73 | −50.60 | 202.37 |
| X—5—5 | 23.00 | 60.00 | −50.00 | 200.00 | 23.92 | 59.47 | −49.55 | 198.17 |
| 0—Y—0 | 22.00 | 60.00 | −50.00 | 200.00 | 20.16 | 60.52 | −50.42 | 201.66 |
| 0—Y—5 | 22.00 | 60.00 | −50.00 | 200.00 | 21.82 | 60.20 | −50.15 | 200.59 |
| 5—Y—0 | 25.00 | 60.00 | −50.00 | 200.00 | 23.24 | 60.68 | −50.55 | 202.19 |
| 5—Y—5 | 25.00 | 60.00 | −50.00 | 200.00 | 23.32 | 60.61 | −50.49 | 201.95 |
| 0—0—Z | 25.00 | 60.00 | −50.00 | 200.00 | 25.11 | 59.78 | −49.81 | 199.20 |
| 0—5—Z | 25.00 | 60.00 | −50.00 | 200.00 | 24.76 | 60.01 | −50.00 | 199.97 |
| 5—0—Z | 25.00 | 60.00 | −50.00 | 200.00 | 25.31 | 59.79 | −49.81 | 199.24 |
| 5—5—Z | 25.00 | 60.00 | −50.00 | 200.00 | 25.20 | 59.95 | −49.95 | 199.76 |

**Table 9.** Comparison between actual values and computed results at 100 mm.

| Combination Sets | Calculated Value | | | | Simulated Value | | | |
| | Angle (°) | Coordinate (mm) | | | Angle (°) | Coordinate (mm) | | |
| | | *x* | *y* | *z* | | *x* | *y* | *z* |
|---|---|---|---|---|---|---|---|---|
| X—0—0 | 12.00 | 60.00 | −50.00 | 100.00 | 10.74 | 60.12 | −50.12 | 100.23 |
| X—0—5 | 13.00 | 60.00 | −50.00 | 100.00 | 11.23 | 59.96 | −49.99 | 99.97 |
| X—5—5 | 25.00 | 60.00 | −50.00 | 100.00 | 23.13 | 60.66 | −50.57 | 101.13 |
| X—5—5 | 25.00 | 60.00 | −50.00 | 100.00 | 23.37 | 60.67 | −50.58 | 101.15 |
| 0—Y—0 | 25.00 | 60.00 | −50.00 | 100.00 | 23.20 | 60.48 | −50.42 | 100.82 |
| 0—Y—5 | 25.00 | 60.00 | −50.00 | 100.00 | 23.15 | 60.63 | −50.54 | 101.08 |
| 5—Y—0 | 25.00 | 60.00 | −50.00 | 100.00 | 23.55 | 60.48 | −50.42 | 100.83 |
| 5—Y—5 | 25.00 | 60.00 | −50.00 | 100.00 | 23.72 | 60.51 | −50.44 | 100.87 |
| 0—0—Z | 25.00 | 60.00 | −50.00 | 100.00 | 25.09 | 59.96 | −49.98 | 99.96 |
| 0—5—Z | 25.00 | 60.00 | −50.00 | 100.00 | 24.52 | 59.97 | −50.00 | 99.98 |
| 5—0—Z | 25.00 | 60.00 | −50.00 | 100.00 | 25.30 | 59.94 | −49.97 | 99.92 |
| 5—5—Z | 25.00 | 60.00 | −50.00 | 100.00 | 25.02 | 60.02 | −50.04 | 100.06 |

### 9.2. Convergence Area of the Simulation Results

Table 10 shows the percentages of the area that converged for the five working distances ranging from 500 mm to 100 mm, calculated from the 2D graph. It shows that the area of convergence increases with reduced working distance z in most rotation combination sets. However, the area decreases with further translation for rotation combinations X—0—0, X—0—5, 0—Y—0, and 0—Y—5. Overall, the simulation shows 77.28% of the converged results.

**Table 10.** Convergence area of the simulation results.

| Combination Sets | Convergence Percentage (%) at Working Distance | | | | |
| | 500 mm | 400 mm | 300 mm | 200 mm | 100 mm |
|---|---|---|---|---|---|
| X—0—0 | 88.21 | 86.39 | 78.23 | 78.00 | 74.60 |
| X—0—5 | 94.33 | 93.42 | 83.90 | 78.46 | 75.96 |
| X—5—0 | 55.10 | 55.56 | 56.92 | 58.05 | 68.03 |
| X—5—5 | 56.69 | 55.78 | 57.60 | 60.54 | 68.03 |
| 0—Y—0 | 79.14 | 82.99 | 76.87 | 77.78 | 74.60 |
| 0—Y—5 | 86.17 | 90.48 | 76.87 | 75.28 | 72.34 |
| 5—Y—0 | 54.65 | 54.88 | 55.10 | 57.37 | 67.35 |
| 5—Y—5 | 53.29 | 53.51 | 54.20 | 57.37 | 100.00 |
| 0—0—Z | 100.00 | 100.00 | 100.00 | 100.00 | 100.00 |
| 0—5—Z | 88.89 | 86.17 | 86.39 | 91.38 | 100.00 |
| 5—0—Z | 88.89 | 87.07 | 84.13 | 85.26 | 98.19 |
| 5—5—Z | 77.32 | 77.78 | 78.00 | 86.85 | 96.37 |

### 9.3. Discussions and Comments

Overall, the area of the converged results and the maximum angle of rotation that is achievable increase when the QR marker's distance moves nearer to the camera. This is because when the QR marker is further away, the changes in rotation angle are barely noticeable as the amount of rotation is analogous to the working distance. The countable rotation angle is inversely proportional to the working distance. In other words, when the working distance is nearer, the distortion caused by the orientation of the QR marker is more noticeable. Additionally, for the rotation combination sets that involve two or more axes, the tilting and distortion of the QR marker are much more complicated, which leads to certain areas showing diverged results.

For combination sets at the maximum working distance of 500 mm away from the camera, which involve rotation around only one single axis or the z-axis together with either the x- or y-axis, more than 75% of the area of the simulated coordinate plane shows converged results. Additionally, among the combination sets for rotation around the x-axis,

the combination with the 0° y-axis and 5° z-axis rotation shows the best result for 94.33% of the converged area. The rotation around the y-axis with a 5° x-axis and 5° z-axis rotated combination shows that only about 53% of the area can yield an angle of rotation and distance within the error tolerance range. Furthermore, the simulation results of different combinations of rotation around the z-axis show that the z-axis rotation has a minor effect on the computation, as 77% of the area of the field of view converged. The positional information can be calculated accurately at all points for rotation around the z-axis and without rotation from other axes. Nevertheless, the computation convergence area shrank to around 53% with the combination sets that included both x- and y-axis rotation. The feasibility of the proposed monocular-vision-based positioning system using a QR code as the landmark via numerical method computation has been verified and is ready for real-world verification.

### 9.4. Comparison between the Proposed Method with Previous Work

A point-to-point comparison is made with the information extract from the graphical data presented in [19]. The data are extracted from the graphs with $\mp0.2$ mm tolerance. The absolute error levels of computed x-, y-, and z-axis locations with 25° z-axis rotation and 60 mm translation in the x-axis are compared and listed in Table 11. The proposed positioning method achieved greater accuracy with less than 3 mm error. Although the error rate of the proposed model was lower compared to [19], the convergence rate for the proposed method was not 100%. However, the method used by [19] can perform positioning at highly translated locations with satisfactory performance.

**Table 11.** Comparison of the absolute error levels on the x-, y-, and z-axis translations affected by various factors.

| Factors | Absolute Error in Previous Work (mm) [18] | | | Absolute Error in Proposed Method (mm) | | |
|---|---|---|---|---|---|---|
| | x-Axis | y-Axis | z-Axis | x-Axis | y-Axis | z-Axis |
| 25° z-axis rotation | 5.0 | 10.0 | 5.0 | 0.4 | 0.3 | 2.3 |
| 60 mm × translation | 10.4 | 6.0 | 6.0 | 0.4 | 0.3 | 0.7 |

## 10. Conclusions and Future Works

The proposed lightweight numerical-based positioning model successfully simplifies the extraction of positional information from one single 2D image. The proposed model applies the pinhole imaging theory and similar triangles rule to find the geometric relationship between the QR code and the captured image. Then, a positioning model is developed using the numerical method to estimate the QR code's depth information and to perform computations toward convergence using the updating rule. The four corner points of the QR code are the positioning points for the model. The 3D coordinates can be identified using the 2D image coordinates and the guessing parameters from the underdetermined system. In addition, the QR code's orientation in the 3D environment is calculated using an inversed rotation matrix. Next, data verification for 12 rotation combination sets around the three cardinal axes is performed using the MATLAB computing platform. The maximum rotation angle at various coordinates at five working distances from 100 mm to 500 mm is determined. A comparison between the calculated and simulated results is accomplished for the 3D position and orientation information with error levels of less than two degree and 5 mm within 30 iterations. Overall, 77.28% of the simulation results converged. The feasibility of the proposed monocular vision-based positioning system using a QR code as the landmark via a numerical method of computation has been verified via computations and is ready for real-world verification.

Our future research work will involve experimental verification for the positioning system. The hardware specifications will be synchronized with the application requirements. Then, an experimental prototype will be built using a high-speed computation processor, pinhole-type cameras, and QR codes available on the market. Experiments will be carried

out using the same parameters and variables as for the simulated model to compare the similarities. A difference between the simulated and experimental results of not more than 15% is tolerable to proceed with real-life applications. Research and comparisons are required to ensure the hardware chosen is sufficient to accommodate the system's requirements. At the same time, the communication between the camera and robot control system will be established for the processor to obtain the images and provide the position and orientation information to the robot control system. Lastly, the experimentally verified system will be implemented in a robot grasping automation system to perform real-time pick and place operations.

**Author Contributions:** Conceptualization, M.K.T. and H.P.Y.; methodology, H.P.Y. and M.K.T.; software, M.K.T.; validation, M.K.T. and H.P.Y.; formal analysis, M.K.T.; investigation, M.K.T.; resources, H.P.Y.; data curation, M.K.T. and H.P.Y.; writing—original draft preparation, M.K.T.; writing—review and editing, H.P.Y., M.K.T. and K.T.K.T.; visualization, M.K.T.; supervision, H.P.Y. and K.T.K.T.; project administration, H.P.Y. and K.T.K.T.; funding acquisition, H.P.Y. and K.T.K.T. All authors have read and agreed to the published version of the manuscript.

**Funding:** This research was funded by Universiti Malaysia Sabah, Special Funding Scheme (SDK0216-2020).

**Institutional Review Board Statement:** Not applicable.

**Informed Consent Statement:** Not applicable.

**Data Availability Statement:** Not applicable.

**Conflicts of Interest:** The authors declare no conflict of interest. The funders had no role in the design of the study; in the collection, analyses, or interpretation of data; in the writing of the manuscript, or in the decision to publish the results.

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
