# Peer review of "Numerical Computation-Based Position Estimation for QR Code Object Marker: Mathematical Model and Simulation"

_computation, doi:10.3390/computation10090147_

Round 1

Reviewer 1 Report

Comments:

The authors have investigated the ‘Numerical Computation-Based Position Estimation for QR Code Object Marker: Mathematical Model and Simulation’. Even though the subject matter is interesting; however, this article has not been written systematically and hence it suffers from significant shortcomings and some of them are listed below:

11. Usually, an abstract of a reader-friendly, scientific article is the ‘self-dependent and concise summary’ of the whole investigation; this is a MANDATORY criterion for writing an abstract. Unfortunately, the current version of the abstract does not satisfy the criterion, as stated above. So, a reader-friendly abstract MUST be written in such a way so that the potential readers, regardless their discipline or expertise, can easily find the following information systematically and easily for example ‘the definition of the problem of investigation’, ‘the precise description of applied methodologies’, and ‘the key findings of the investigation, which are unique and universally valid under the wide range of pertinent conditions.  This is one of the major shortcomings of this article.

Besides, some unnecessary sentences/phrases are included in the abstract, which may be appropriate elsewhere, for instance, in the introduction part.   Hence it is recommended to revise the abstract as suggested above.       

22. The next important and most vital part of an article is the ‘introduction’. It is generally treated as the heart of an article. The introduction part usually guides the flow of the construction of the rest of the article’s parts. So, a question naturally arises on how to construct the introduction of a reader-friendly scientific article? To address the above question, the authors are strongly suggested here to survey the existing literature on the subject matter of the article extensively to reveal the ‘research gap or originality’ within the existing literature. Once the ‘research gap’ has been identified, then the rest of the article MUST be devoted to filling up the ‘research gap’ as identified. Unfortunately, the ‘introduction’ of this article is not written as highlighted above. In other words, the authors have completely failed to reveal the ‘research gap’ of this investigation. Without revealing the ‘research gap’ systematically, any research work has no scientific value! The ‘research gap’ MUST be revealed systematically and it should have both physical and scientific reasons for writing any ‘research gap’, which is again an obligatory event!  Hence, extensive modification of ‘introduction’ is mandatory event for this article.

3. Since this work has been completed by adopting MATLAB, a separate section MUST be added for ‘applied boundary conditions’ just after the description of the mathematical model, that is, governing equations.  It is a mandatory event for the article which is the output of a numerical study.

 4. The  quality of the Figs that are presented in this article is pretty poor. It is noticed here that no post-processing of Figs. is conducted!! Figures should not contain any part of the window information of MATLAB.  Just copping Figs from MATLAB and pasting those Figs in the article won't work to produce good quality Figs for a scientific article. Hence, it is recommended that the post-processing and re-drawing of Figs obtained by MATLAB is necessary.

55. No validation study is found in this article and hence it is another drawback of this article. Furthermore, it is a common practice of writing any kind of article adopting numerical study. Hence, it is imperative to find some way to validate the results that are presented in this article. However, the presented result is suffering from reliability problems!! So, the authors MUST address this event in a systematic and scientific fashion. 

66. Finally, the conclusion part MUST also be precise and straightforward as an abstract so that the potential readers can easily understand events as mentioned above (1) along with major findings of the article. Unfortunately, the ‘conclusion’ part is not written as expected. Besides, the ‘conclusion’ must have consistency with the abstract; this is a common practice of writing a reader-friendly scientific article. Hence, a revision is necessary as suggested above.

77. This article has other minor problems which MUST be detected by the authors and hence try to address them accordingly. Hence, reviewers would like to see the overall construction of this article with the aim of filling the ‘research gap’ as mentioned earlier.  Hence, a major revision is necessary.

 Anyway, please wait for the comments from the editor.

Author Response

Comments Response Report

Manuscript ID: computation-1826087

The authors highly appreciate the reviewer’s effort for their helpful comments and carefulness review on the paper. The revised version of the paper had addressed all the comments and made good. The detailed responses to the comment are as follow.

Response to Reviewer 1 Comments

  • Usually, an abstract of a reader-friendly, scientific article is the ‘self-dependent and concise summary’ of the whole investigation; this is a MANDATORY criterion for writing an abstract. Unfortunately, the current version of the abstract does not satisfy the criterion, as stated above. So, a reader-friendly abstract MUST be written in such a way so that the potential readers, regardless their discipline or expertise, can easily find the following information systematically and easily for example ‘the definition of the problem of investigation’, ‘the precise description of applied methodologies’, and ‘the key findings of the investigation, which are unique and universally valid under the wide range of pertinent conditions. This is one of the major shortcomings of this article. Besides, some unnecessary sentences/phrases are included in the abstract, which may be appropriate elsewhere, for instance, in the introduction part. Hence it is recommended to revise the abstract as suggested above.

  • The abstract has been revised, and we removed the unnecessary sentences. Besides, we includes the research problem, concepts used to develop the model and key findings of the simulation as suggested.

  • The next important and most vital part of an article is the ‘introduction’. It is generally treated as the heart of an article. The introduction part usually guides the flow of the construction of the rest of the article’s parts. So, a question naturally arises on how to construct the introduction of a reader-friendly scientific article? To address the above question, the authors are strongly suggested here to survey the existing literature on the subject matter of the article extensively to reveal the ‘research gap or originality’ within the existing literature. Once the ‘research gap’ has been identified, then the rest of the article MUST be devoted to filling up the ‘research gap’ as identified. Unfortunately, the ‘introduction’ of this article is not written as highlighted above. In other words, the authors have completely failed to reveal the ‘research gap’ of this investigation. Without revealing the ‘research gap’ systematically, any research work has no scientific value! The ‘research gap’ MUST be revealed systematically and it should have both physical and scientific reasons for writing any ‘research gap’, which is again an obligatory event! Hence, extensive modification of ‘introduction’ is mandatory event for this article.

  • The literature review is restructured to systematically present the research. Besides, the advantages of proposed method compared to the current system is also discussed. Please refer to the introduction section:

Research gap is explained in first paragraph of introduction.

Some limitations of the literature reviewed discusses in the second and third paragraph.

Forth paragraph discusses the advantages and contribution of the proposed system.

Section 7 presents the discussion and comparison with the similar work. The advantages of the proposed method is discussed.

  • Since this work has been completed by adopting MATLAB, a separate section MUST be added for ‘applied boundary conditions’ just after the description of the mathematical model, that is, governing equations. It is a mandatory event for the article which is the output of a numerical study.

  • Section 8 ‘ Positioning model simulation using MATLAB’ is presented. Explanation for the workflow, parameters and variables used for the simulation are included.

  • The quality of the Figs that are presented in this article is pretty poor. It is noticed here that no post-processing of Figs. Is conducted!! Figures should not contain any part of the window information of MATLAB. Just copping Figs from MATLAB and pasting those Figs in the article won't work to produce good quality Figs for a scientific article. Hence, it is recommended that the post-processing and re-drawing of Figs obtained by MATLAB is necessary.

  • Post-processing of Figs by removing the redundant window information is completed. However, re-drawing of the Figure is not necessary as the Figures illustrate the result in correct way; to note the maximum rotation angle detectable at each points in 2D form, and visualise the area with converged simulation.

  • No validation study is found in this article and hence it is another drawback of this article. Furthermore, it is a common practice of writing any kind of article adopting numerical study. Hence, it is imperative to find some way to validate the results that are presented in this article. However, the presented result is suffering from reliability problems!! So, the authors MUST address this event in a systematic and scientific fashion.

  • Result validation subsection is included. The comparison between calculated and simulated value is presented. The simulation results is validated and listed from Table 5 to Table 9.

  • Finally, the conclusion part MUST also be precise and straightforward as an abstract so that the potential readers can easily understand events as mentioned above (1) along with major findings of the article. Unfortunately, the ‘conclusion’ part is not written as expected. Besides, the ‘conclusion’ must have consistency with the abstract; this is a common practice of writing a reader-friendly scientific article. Hence, a revision is necessary as suggested above.

  • The conclusion is revised and key findings are included. Besides, the scope for the future work are discussed.

  • This article has other minor problems which MUST be detected by the authors and hence try to address them accordingly. Hence, reviewers would like to see the overall construction of this article with the aim of filling the ‘research gap’ as mentioned earlier. Hence, a major revision is necessary.

  • The articles is revised and make good. Introduction section included the explanation of the research gap and how the proposed method can overcome the limitation. Besides, the advantages of proposed approach is also discussed in details. Please refer to section 7 for the discussion and comparison made with the similar research found.

Reviewer 2 Report

*       The significant trends of the simulation results should show.

*       Comparison with recent studies and methods would be appreciated.

*       Introduction section can add the issues in the current work context and how proposed algorithms/approaches can overcome this.

*       Literature review techniques have to be strengthened by including the current system's issues and how the author proposes to overcome the same.

*       Clarify the finding Error rate and accuracy in the performance analysis section.

*       It is suggested to add the chart for the given process with a description.

*       The mapping process for the proposed technique should be discussed in detail.

*       Conclusion should state scope for future work.

*       Authors should add more information on the code's implementation to perform the analysis and the library involved in this task.

*       Authors should add the parameters of the process/method.

*       The paper does not clearly explain its advantages concerning the literature: the novelty and contributions of the proposed work are not clear: does it offer a new method? Or does the innovation only consists of the application?

The advantage of the proposed method concerning other ways in the literature should be clarified.

Author Response

Comments Response Report

Manuscript ID: computation-1826087

The authors highly appreciate the reviewer’s effort for their helpful comments and carefulness review on the paper. The revised version of the paper had addressed all the comments and made good. The detailed responses to the comment are as follow.

Response to Reviewer 2 Comments

Point 1: The significant trends of the simulation results should show.

Response 1: Subsection 9.3 and 9.4 presents the key finding from the simulation results.

Point 2: Comparison with recent studies and methods would be appreciated.

Response 2: Section 7 ‘Comments on the Proposed Method and Comparison with Previous Work’ discusses about the similar studies and the method used. Comparison is made and the advantages of proposed method is presented. For results part, point-to-point comparision with previous work is made and the absolute error comparison are listed in Table 11 in subsection 9.5.

Point 3: Introduction section can add the issues in the current work context and how proposed algorithms/approaches can overcome this.

Response 3: The research gap identified and existing issues is presented in Introduction part. The proposed approach to overcome the limitations is also discussed. Please read the introduction section’s first and forth paragraph for more details.

Point 4: Literature review techniques have to be strengthened by including the current system's issues and how the author proposes to overcome the same.

Response 4: The literature review is restructured to systematially present the research. Besides, the advantages of proposed method compared to the current system is also discussed. Please refer to the introduction section:

The limitations of the literature reviewed is added in second and third paragraph in the introduction.

Forth paragraph discusses the advantages and contribution of the proposed system.

The positioning system can overcome the limitations found (to the best of our knowledge) in the current work. Comparison with the similar research is also presented in Section 7 and subsection 9.5.

Point 5: Clarify the finding Error rate and accuracy in the performance analysis section.

Response 5: The details of convergence percentage for the simulation results is presented in Subsection 9.3 and 9.4.

Point 6: It is suggested to add the chart for the given process with a description.

Response 6: The chart to show the ‘main workflow for mathematical model’ is added to better shows the process to develop the positioning model and related concepts for each step:

Figure 2 shows the chart for the main workflow of the positioning model with explanation in the first paragraph of section 3.

Figure 7 in section 8 shows the workflow for the simulation process. Each steps with related equation is included and discussed.

Point 7: The mapping process for the proposed technique should be discussed in detail.

Response 7: Please refer to Equation 3 to Equation 4 for the mapping process. The x- and y-coordinate of a point in 3D environment is map to the 2D image using Equation 4 and 5. The general form of the mapping equation is found in Equation 6. The proposed method then use numerical method to guess the value of z coordinate and thus determine the x and y coordinate.

Point 8: Conclusion should state scope for future work.

Response 8: The conclusion had been revised and the scope for future works are included.

Point 9: Authors should add more information on the code's implementation to perform the analysis and the library involved in this task.

Response 9: Explanation and details related to the simulation process is discussed in Section 8 ‘ Positioning Model Simulation Using MATLAB’. The workflow and description are presented. However, the list of library is not presented as the code is fully implemented by the author, no library is used the simulation model is develop using basic MATLAB command.

Point 10: Authors should add the parameters of the process/method.

Response 10: Explanation and details related to the simulation process is discussed in Section 8 ‘ Positioning Model Simulation Using MATLAB’. The parameters and variables for the simulation are presented.

Point 11: The paper does not clearly explain its advantages concerning the literature: the novelty and contributions of the proposed work are not clear: does it offer a new method? Or does the innovation only consists of the application?

The advantage of the proposed method concerning other ways in the literature should be clarified.

Response 11: The advantages and mian contributions of proposed method are presented in the forth paragraph of introduction section. In addition, the section 7 ‘Comments on the Proposed Method and Comparison with Previous Work’ discusses and compares the advantages of proposed method. The comparison of the result of the proposed method and existing work is presented in subsection 9.5.

Section 7 ‘Comments on the Proposed Method and Comparison with Previous Work’ discusses about the similar studies and the method used. Comparison is made and the advantages of proposed method is presented. For results part, point-to-point comparision with previous work is made and the absolute error comparison are listed in Table 11 in subsection 9.5.

Point 12: Introduction section can add the issues in the current work context and how proposed algorithms/approaches can overcome this.

Response 12: The research gap identified and existing issues is presented in Introduction part. The proposed approach to overcome the limitations is also discussed. Please read the introduction section’s first and forth paragraph for more details.

Point 13: Literature review techniques have to be strengthened by including the current system's issues and how the author proposes to overcome the same.

Response 13: The literature review is restructured to systematially present the research. Besides, the advantages of proposed method compared to the current system is also discussed. Please refer to the introduction section:

The limitations of the literature reviewed is added in second and third paragraph in the introduction.

Forth paragraph discusses the advantages and contribution of the proposed system.

The positioning system can overcome the limitations found (to the best of our knowledge) in the current work. Comparison with the similar research is also presented in Section 7 and subsection 9.5.

Point 14: Clarify the finding Error rate and accuracy in the performance analysis section.

Response 14: The details of convergence percentage for the simulation results is presented in Subsection 9.3 and 9.4.

Point 15: It is suggested to add the chart for the given process with a description.

Response 15: The chart to show the ‘main workflow for mathematical model’ is added to better shows the process to develop the positioning model and related concepts for each step:

Figure 2 shows the chart for the main workflow of the positioning model with explanation in the first paragraph of section 3.

Figure 7 in section 8 shows the workflow for the simulation process. Each steps with related equation is included and discussed.

Point 16: The mapping process for the proposed technique should be discussed in detail.

Response 16: Please refer to Equation 3 to Equation 4 for the mapping process. The x- and y-coordinate of a point in 3D environment is map to the 2D image using Equation 4 and 5. The general form of the mapping equation is found in Equation 6. The proposed method then use numerical method to guess the value of z coordinate and thus determine the x and y coordinate.

Point 17: Conclusion should state scope for future work.

Response 17: The conclusion had been revised and the scope for future works are included.

Point 18: Authors should add more information on the code's implementation to perform the analysis and the library involved in this task.

Response 18: Explanation and details related to the simulation process is discussed in Section 8 ‘ Positioning Model Simulation Using MATLAB’. The workflow and description are presented. However, the list of library is not presented as the code is fully implemented by the author, no library is used the simulation model is develop using basic MATLAB command.

Point 19: Authors should add the parameters of the process/method.

Response 19: Explanation and details related to the simulation process is discussed in Section 8 ‘ Positioning Model Simulation Using MATLAB’. The parameters and variables for the simulation are presented.

Point 20: The paper does not clearly explain its advantages concerning the literature: the novelty and contributions of the proposed work are not clear: does it offer a new method? Or does the innovation only consists of the application?

The advantage of the proposed method concerning other ways in the literature should be clarified.

Response 20: The advantages and mian contributions of proposed method are presented in the forth paragraph of introduction section. In addition, the section 7 ‘Comments on the Proposed Method and Comparison with Previous Work’ discusses and compares the advantages of proposed method. The comparison of the result of the proposed method and existing work is presented in subsection 9.5.

Round 2

Reviewer 1 Report

Please wait for the comments from the editorial  office of 'Computation'